# WEIGHT-ENTANGLEMENT MEETS GRADIENT-BASED NEURAL ARCHITECTURE SEARCH

## ABSTRACT

Weight sharing is a fundamental concept in neural architecture search (NAS), enabling gradient-based methods to explore cell-based architecture spaces significantly faster than traditional blackbox approaches. In parallel, weight *entanglement* has emerged as a technique for intricate parameter sharing among architectures within macro-level search spaces. Since weight-entanglement poses compatibility challenges for gradient-based NAS methods, these two paradigms have largely developed independently in parallel sub-communities. This paper aims to bridge the gap between these sub-communities by proposing a novel scheme to adapt gradient-based methods for weight-entangled spaces. This enables us to conduct an in-depth comparative assessment and analysis of the performance of gradient-based NAS in weight-entangled search spaces. Our findings reveal that this integration of weight-entanglement and gradient-based NAS brings forth the various benefits of gradient-based methods (enhanced performance, improved supernet training properties and superior any-time performance), while preserving the memory efficiency of weight-entangled spaces. The code for our work is openly accessible here.

## 1 INTRODUCTION

The concept of weight-sharing in Neural Architecture Search (NAS) arose from the need to improve the efficiency of conventional blackbox NAS algorithms, which demand significant computational resources to evaluate individual architectures. Here, weight-sharing (WS) refers to the paradigm by which we represent the search space with a single large *supernet*, also known as the *one-shot* model, that subsumes all the candidate architectures in that space. Every edge of this supernet holds all the possible operations that can be assigned to that edge. Importantly, architectures that share a particular operation also share its corresponding operation weights, allowing simultaneous training of an exponential number of subnetworks, unlike the sequential approach of blackbox NAS.

Gradient-based NAS algorithms (or *optimizers*), such as DARTS (Liu et al., 2019), GDAS (Dong and Yang, 2019) and DrNAS (Chen et al., 2021b), assign an *architectural parameter* to every choice of operation on a given edge of the supernet. The output feature maps of these edges are thus an aggregation of the outputs of the individual operations on that edge, weighted by their architectural parameters. These architectural parameters are learned using gradient updates by differentiating through the validation loss. Supernet weights and architecture parameters are therefore trained simultaneously in a bi-level fashion. Once this training phase is complete, the final architecture can be identified quickly, e.g., by selecting operations with the highest architectural weights on each edge as depicted in Figure 1(b). However, more sophisticated methods have also been explored (Wang et al., 2021) for this selection.

While gradient-based NAS methods have primarily been studied for cell-based NAS search spaces, a different class of search spaces focuses on macro-level structures (parameterizing kernel size, number of channels, etc.) for which all architectures in the space are *subnetworks* of the architecture with the largest architectural choices, which is identical to the supernet in this case. These search spaces share weights via the more intricate form of *weight-entanglement* (WE) between similar operations on the same edge; e.g., the nine weights of a $3 \times 3$ convolution are a subset of the 25 weights of a $5 \times 5$ convolution. This paradigm reduces the memory requirements of the supernet to the size of the largest architecture in the search space.

In order to efficiently search over such weight-entanglement spaces, *two-stage* methods have been introduced that first pre-train the supernet, and then perform blackbox search on it to obtain the final architecture. OFA (Cai et al., 2020), SPOS (Guo et al., 2020), AutoFormer (Chen et al., 2021a) and HAT (Wang et al., 2020) are prominent examples of methods that fall into this category. Note that these methods do not employ additional architectural parameters for supernet training or search. They typically train the supernet by randomly sampling subnetworks and training them as depicted in Figure 1(a). The post-hoc blackbox search relies on using the performance of subnetworks sampled from the trained supernet as a proxy for true performance on the unseen test set. To contrast with this two-stage approach, we refer to traditional gradient-based NAS approaches as *single stage* methods.

While to date, weight-entangled spaces have only been explored with two-stage methods, and cell-based spaces have only been optimized with single-stage approaches, in this paper we bridge the gap between these parallel sub-communities. We do so by addressing the challenges associated with integrating off-the-shelf single-stage NAS methods with weight-entangled search spaces.

After a discussion of related work (Section 2), we make the following main contributions:

- We propose a generalized scheme to apply single-stage methods to weight-entangled spaces, while maintaining search efficiency and efficacy at larger scales (Section 3, with visualizations in Figure 1(c) and Figure 2). We refer to this method as *TangleNAS*.
- We propose a fair playground for the comparative evaluation of single and two-stage methods (Section 4.1) and study the effect of weight-entanglement in conventional cell-based search spaces (i.e., NASBench201 and the DARTS search space) (Section 4.2).
- We evaluate our proposed generalized scheme for single-stage methods across a diverse set of weight-entangled macro search spaces and tasks, from image classification (Section 4.3.1, Section 4.3.2) to language modeling (Section 4.3.3).
- We conduct a comprehensive evaluation of the properties of single and two-stage approaches including any-time-performance, memory consumption, robustness to training fraction and effect of fine-tuning (Section 5), demonstrating that our generalized gradient-based NAS method achieves the best of single and two-stage methods: the enhanced performance, improved supernet fine-tuning properties, superior any-time performance of single-stage methods, and the low memory consumption of two-stage methods .

To facilitate reproducibility, our code is openly accessible here.

## 2 RELATED WORK

*Weight-sharing* was first introduced in ENAS (Pham et al., 2018), which reduced the computational cost of NAS by 1000× compared to previous methods. However, since this method used reinforcement learning its performance was quite brittle. Bender et al. (2018) simplified the technique, showing that searching for good architectures is possible by training the supernet directly with stochastic gradient descent. This was followed by DARTS (Liu et al., 2019), which set the cornerstone for efficient and effective gradient-based, *single-stage*, NAS approaches.

DARTS, however, had prominent *failure modes*, such as its *discretization gap* and *convergence towards parameter-free operations*, as outlined in Robust-DARTS (Zela et al., 2020). Numerous gradient-based one-shot optimization techniques were developed since then (Cai et al., 2019; Nayman et al., 2019; Xu et al., 2019; Dong and Yang, 2019; Hu et al., 2020; Li et al., 2021; Chen et al., 2021b; Zhang et al., 2021). Among these, we highlight DrNAS (Chen et al., 2021b), which we will use in our experiments as a representative of gradient-based NAS methods. DrNAS treats one-shot search as a distribution learning problem, where the parameters of a Dirichlet distribution over architectural parameters are learned to identify promising regions of the search space.

Despite the remarkable performance of single-stage methods, they are not directly applicable to some real-world architectural domains, such as transformers, because of the macro-level structure of these search spaces. DASH (Shen et al., 2022) employs a DARTS-like methodology to optimize CNN topologies (i.e. kernel size, dilation) for a diverse set of tasks, reducing computational complexity by appropriately padding and summing kernels with different sizes and dilations. FBNet-v2 (Wan et al., 2020) makes an attempt along these lines for CNN topologies, but its methodology is not easily

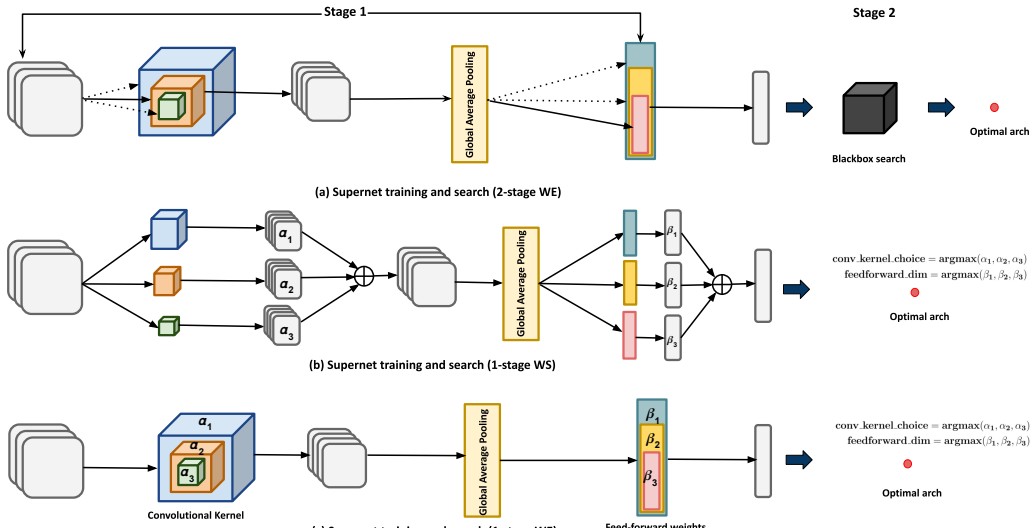

Figure 1: (a) **Two-Stage NAS with WE** (Algorithm 3): dotted paths show operation choices not sampled at the given step (b) **Single-Stage NAS with WS** (Algorithm 4): every operation choice is evaluated independently and contributes to the output feature map with corresponding architecture parameters (c) **Single-Stage NAS with WE** (Algorithm 1): operation choices superimposed with corresponding arhictecture parameters.

extendable to search spaces like transformers with multiple interacting modalities, such as embedding dimension, number of heads, expansion ratio, and depth.

Weight-entanglement, on the other hand, provides a more effective way of weight-sharing exclusive to macro-level architectural spaces. The concept of weight-entanglement was developed in slimmable networks (Yu et al., 2018; Yu and Huang, 2019), OFA (Cai et al., 2020) and BigNAS (Yu et al., 2020) in the context of convolutional networks (see also AtomNAS (Mei et al., 2019)) and later spelled out in AutoFormer (Chen et al., 2021a), where it was applied to the transformer architecture.

Single-path-one-shot (SPOS) methods (Guo et al., 2020) have shown a lot of promise in searching weight-entangled spaces. SPOS trains a supernet by uniformly sampling single paths (one at a time to limit memory consumption) and then training the weights along that path. The supernet training is followed by a black-box search that uses the performance of the models sampled from the trained supernet as a *proxy*.

OFA used a similar idea to optimize different dimensions of CNN architectures, such as its depth, width, kernel size and resolution. Additionally, it enforced training of larger to smaller sub-networks sequentially to prevent interference between sub-networks. Subsequently, AutoFormer adopted the SPOS method to optimize a weight-entangled space of transformer architectures.

In this work, we demonstrate how single-stage methods can be applied to macro-level search spaces with weight-entanglement, where one can benefit from the time efficiency and remarkable effectiveness of modern differentiable NAS optimizers, while concurrently maintaining memory efficiency of the weight-entangled space. While we chose to adopt DrNAS as the primary approach for our exploration of the weight-entangled space in this study, it is worth highlighting that our methodology is generally applicable to other gradient-based NAS methods.

## 3 METHODOLOGY: SINGLE-STAGE NAS WITH WEIGHT-SUPERPOSITION

Our primary goal in this work is to effectively apply single-stage NAS to search spaces with macro-level architectural choices. To reduce the memory consumption and preserve computational efficiency, we propose two major modifications to single-stage methods. In our framework, the operators on any given edge inherit the weights of the largest operator on that edge. This reduces the size of the supernet to the size of the largest individual architecture in the search space.

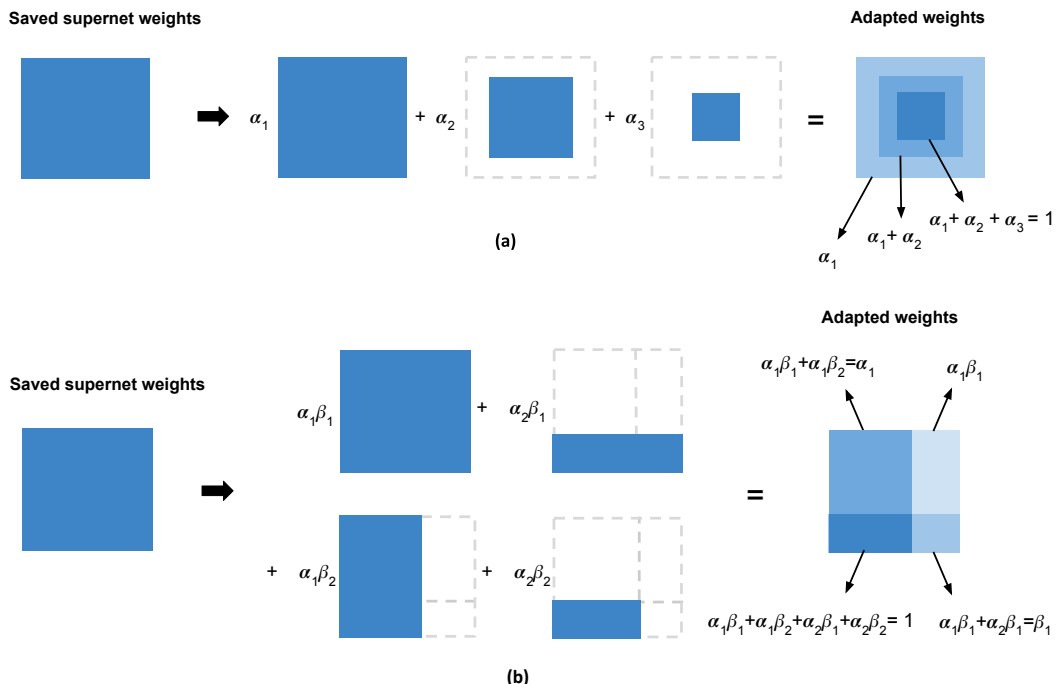

Figure 2: Supernet weight matrix (LHS) adapted to gradient-based methods (RHS). All operation choices are zero-padded to match the dimension of the largest operation and superimposed with given architecture parameter. (a) Weight superposition for a single dimension choice (e.g. kernel size) with architecture parameter $\alpha_i$ (b) Weight superposition for combination of multiple dimensions (e.g. embedding dimension and expansion ratio) with architecture parameters $\alpha_i$, $\beta_j$. White areas inside the dashed gray boundaries indicate zero-padding and color values reflect scaling.

Then, instead of weighting the outputs of operations on a given edge by architectural parameters, as single-stage NAS methods do (see Figure 1(b)), we simply take each operation choice, zero-pad that to match the dimensions of the largest operation and sum these terms, each weighted by the corresponding architectural parameter. Figure 2(a) provides an overview of the idea for a single architectural choice such as the kernel size. This is equivalent to taking the largest operation and re-scaling the weights of each sub-operation by the sum of architectural parameters of operations it is embedded in (see right-most weight matrix in Figure 2(a).

This process results in a *superposition* of operation weights whose structural property is similar to the largest operation. Consequently, a *single* forward pass suffices to capture the effect of all operational choices, thus making our approach computationally efficient. Furthermore, to accommodate for operations depending on two or more architectural dimension choices we introduce the combi-superposition outlined in Figure 2(b) and Algorithm 2. This allows us to apply any arbitrary gradient-based NAS method, such as DARTS, GDAS or DrNAS, to macro-level search spaces that leverage weight-entanglement, without additional memory consumption and computational cost during forward propagation. We refer to our memory and compute optimal single-stage architecture search method as *TangleNAS* and provide an overview of the ap-

---

**Algorithm 1** TangleNAS

1: **Input:** $M \leftarrow$ number of cells, $N \leftarrow$ number of operations
   $\mathcal{O} \leftarrow [o_1, o_2, o_3, ...o_N]$
   $\mathcal{W}_{\max} \leftarrow \cup_{i-1}^N w_i$
   $\mathcal{A} \leftarrow [\alpha_1, \alpha_2, \alpha_3, ...\alpha_N]$
   $\gamma =$ learning rate of $\mathcal{A}$, $\eta =$ learning rate of $\mathcal{W}_{\max}$
   $f$ is a function or distribution s.t. $\sum_{i=1}^N f(\alpha_i) = 1$
2: $Cell_j \leftarrow DAG(\mathcal{O}_j, \mathcal{W}_{max\,j})$ /* defined for j=1...M */
3: $Supernet \leftarrow \cup_j^M Cell_j \cup \mathcal{A}$
4: /* example of forward propagation on the cell */
5: **for** $j \leftarrow 1$ to $M$ **do**
6:    /* PAD weight to output dimension of $\mathcal{W}_{\max}$ before summation */
7:    /* Generalized Weighing Scheme */
8:    $\overline{o_j(x, \mathcal{W}_{\max})} = o_{j,i}(x, \sum_{i=1}^N f(\alpha_i)\mathcal{W}_{\max}[: i])$
9: **end for**
10: /* weights and architecture update */
11: $\mathcal{A} = \mathcal{A} - \gamma \nabla_{\mathcal{A}} \mathcal{L}_{val}(\mathcal{W}_{\max}{}^*, \mathcal{A})$
12: $\mathcal{W}_{\max} = \mathcal{W}_{\max} - \eta \nabla_{\mathcal{W}_{\max}} \mathcal{L}_{train}(\mathcal{W}_{\max}, \mathcal{A})$
13: /* Architecture Selection */
14: $selected\_arch \leftarrow \arg\max(\mathcal{A})$

| Search Type | Optimizer | Supernet type | Test acc | Search Time(hrs) |
|---|---|---|---|---|
| Single-Stage | DrNAS | WS | $91.190 \pm 0.049$ | 6.277 |
| | TangleNAS | WE | $\mathbf{91.300 \pm 0.023}$ | 6.222 |
| Two-Stage | SPOS+RS | WE | $90.680 \pm 0.253$ | 15.611 |
| | SPOS+ES | WE | $90.317 \pm 0.223$ | 13.244 |
| Optimum | - | - | 91.630 | - |

Table 1: Evaluation on the toy cell-based search space on the Fashion-MNIST dataset

| Search Type | Optimizer | Supernet type | Test acc | Search Time(hrs) |
|---|---|---|---|---|
| Single-Stage | DrNAS | WS | $10 \pm 0.00$ | 12.361 |
| | TangleNAS | WE | $\mathbf{82.495 \pm .0.461}$ | 8.55 |
| Two-Stage | SPOS+RS | WE | $81.253 \pm 0.6717$ | 21.676 |
| | SPOS+ES | WE | $81.890 \pm 0.800$ | 26.359 |
| Optimum | - | - | 84.410 | - |

Table 2: Evaluation on the toy conv-macro search space on the CIFAR10 dataset. Note that DrNAS with weight sharing converges to a degenerate architecture here.

proach in Algorithm 1. The operation $f$ in Algorithm 1 determines the differentiable optimizer used in the method. A $softmax$ function corresponds to DARTS, while sampling from the Dirichlet distribution corresponds to DrNAS. In all experiments of the upcoming sections we use DrNAS as a basic gradient-based NAS method. We therefore simply refer to this combination as TangleNAS throughout the paper.

## 4 EXPERIMENTS

We evaluate TangleNAS across a broad range of search spaces, ranging from cell-based spaces, which serve as the foundation for single-stage methods, to weight-entangled convolutional and transformer spaces, which are central to two-stage methods. We initiate our studies by exploring three simple toy search spaces, which include a collection of cell-based and weight-entangled spaces. Later, we scale our experiments to larger analogs of these spaces. In all our experiments, we use *WE* to refer to the supernet type with entangled weights between operation choices and *WS* to refer to standard weight-sharing proposed in cell-based spaces. For details on our experimental setup please refer to Appendix E. Furthermore, in all our experiments the focus is on *unconstrained search*, i.e., a scenario where the user is interested in obtaining the architecture with the best performance on their metric of choice. The two-stage baselines we mainly compare against are SPOS (Guo et al., 2020) with Random Search (*SPOS+RS*) and SPOS with Evolutionary Search (*SPOS+ES*). For MobileNetV3 (Section 4.3.2) and ViT (Section 4.3.1) we use the original training scheme from OFA (Cai et al., 2020) and Autoformer (Chen et al., 2021a), respectively. Both of these works use SPOS (Guo et al., 2020) as their foundation.

### 4.1 TOY SEARCH SPACES

We begin the evaluation of TangleNAS on two relatively compact *toy* search spaces that we designed as a contribution to the community to allow faster iterations of algorithm development:

- **Toy cell space**: a small version of the DARTS space; architectures are evaluated on the Fashion-MNIST dataset.
- **Toy conv-macro space**: a small space inspired by MobileNet, including kernel sizes and the number of channels in each convolution layer; architectures are evaluated on CIFAR-10.

We describe these spaces in Appendix C, including links to code for these open source toy benchmarks. The results of these experiments are summarized in Tables 1 and 2. Across both of these search spaces, TangleNAS outperforms its *two-stage* counterparts. Also, DrNAS without weight entanglement performs very poorly on the macro level search space.

### 4.2 CELL-BASED SEARCH SPACES

We now start our comparative analysis of single and two-stage approaches by applying them to cell-based spaces that serve as focal points in the single-stage NAS literature to evaluate TangleNAS against DrNAS and SPOS. Here, we use the popular NAS-Bench-201 (NB201) (Dong and Yang, 2020) and DARTS (Liu et al., 2019) search spaces. We refer the reader to Appendix C for details about these spaces and Appendix E.4 for the experimental setup.

Our contribution on these spaces is two-fold. First, we study the effects of weight-entanglement on cell-based spaces in conjunction with single-stage methods. To this end, we entangle the weights of similar operations with different kernel sizes on both search spaces. For NB201, the weights of the 1x1 and 3x3 convolutions are entangled, and in the DARTS search space, the weights of dilated and separable convolutions with kernel sizes 3x3 and 5x5 are entangled. Second, we study SPOS on the NB201 and DARTS search spaces. To the best of our knowledge, we are the first to study a two-stage method like SPOS in such cell search spaces.

Tables 3 and 4 show the results. For both search spaces, TangleNAS yields the best results, outperforming the single-stage baseline DrNAS with WS, as well as both SPOS variants. TangleNAS also significantly lowers the memory requirements and runtime compared to its weight-sharing counterpart. We note that overall, the SPOS methods are ineffective in these cell search spaces.

| Search Type | Optimizer | Supernet | CIFAR10 | CIFAR100 | ImageNet16-120 | Search Time (hrs) |
|---|---|---|---|---|---|---|
| Single-Stage | DrNAS | WS | $94.36 \pm 0.00$ | $72.245 \pm 0.732$ | $46.37 \pm 0.00$ | 20.9166 |
| | TangleNAS | WE | $94.36 \pm 0.00$ | $73.51 \pm 0.000$ | $46.37 \pm 0.00$ | 20.3611 |
| Two-Stage | SPOS+RS | WE | $89.107 \pm 0.884$ | $56.865 \pm 2.597$ | $31.665 \pm 1.1460$ | 29.51388 |
| | SPOS+ES | WE | $87.133 \pm 2.605$ | $56.463 \pm 2.342$ | $29.785 \pm 3.0149$ | 26.74721 |

Table 3: Comparison of test-accuracies of single and two-stage methods on NB201 search space

| Search Type | Optimizer | Supernet | CIFAR10 | ImageNet | Search Time(hrs) |
|---|---|---|---|---|---|
| Single-Stage | DrNAS | WS | $2.625 \pm 0.075$ | 26.29 | 9.12 |
| | TangleNAS | WE | $2.556 \pm 0.034$ | 25.691 | 7.44 |
| Two-Stage | SPOS+RS | WE | $2.965 \pm 0.072$ | 27.114 | 18.7444 |
| | SPOS+ES | WE | $3.200 \pm 0.065$ | 27.320 | 14.794 |

Table 4: Comparison of test errors of single and two-stage methods on the DARTS search space

## 4.3 MACRO SEARCH SPACES

Given the promising results of TangleNAS on the toy and cell-based spaces, we now extend our evaluation to the home base of two-stage methods. We study TangleNAS on a vision transformer space (AutoFormer-T) and a convolutional space (MobileNetV3), which have been proposed and explored using two-stage methods by Chen et al. (2021a) and Cai et al. (2020), respectively, as well as a language model transformer search space built around GPT-2 Radford et al. (2019).

### 4.3.1 AUTOFORMER

| Search Type | Optimizer | CIFAR10 | | | | | CIFAR100 | | | | |
|---|---|---|---|---|---|---|---|---|---|---|---|
| | | Inherit | Fine-tune | Retrain | Params | FLOPS | Inherit | Fine-tune | Retrain | Params | FLOPS |
| Single-Stage | TangleNAS | 93.57 | $97.702 \pm 0.017$ | $97.872 \pm 0.054$ | 8.6847 | 1.946 | 76.59 | $82.615 \pm 0.064$ | $82.668 \pm 0.161$ | 8.64874M | 1.9388G |
| Two-Stage | SPOS+RS | 94.29 | $97.605 \pm 0.038$ | $97.767 \pm 0.024$ | 8.51161M | 1.91G | 78.210 | $82.407 \pm 0.026$ | $82.210 \pm 0.14242$ | 8.47558M | 1.9046G |
| | SPOS+ES | 94.10 | $97.632 \pm 0.047$ | $97.6425 \pm 0.023$ | 7.230286 | 1.6595 | 77.97 | $82.517 \pm 0.140$ | $82.5175 \pm 0.114$ | 8.2447M | 1.859G |

Table 5: Evaluation on the AutoFormer-T space for CIFAR10 and CIFAR100.

We evaluate TangleNAS on the AutoFormer-T space introduced by Chen et al. (2021a), which is based on vision transformers. The search space consists of the choices of embedding dimensions and number of layers, and for each layer, its MLP expansion ratio and the number of heads it uses to compute attention. More details can be found in Table 17 in the appendix. The embedding dimension choice is held constant throughout the network while the number of heads and the MLP expansion ratio varies for every layer. This amounts to a total search space size of about $10^{13}$ architectures. We train our supernet using the same training hyperparameters as used in AutoFormer.

| NAS Method | SuperNet-Type | ImageNet | Datasets | | | | | Params | FLOPS |
|---|---|---|---|---|---|---|---|---|---|
| | | | CIFAR10 | CIFAR100 | Flowers | Pets | Cars | | |
| SPOS+ES | AutoFormer-T | 75.474 | 98.019 | 86.369 | **98.066** | 91.558 | 91.935 | 5.893M | 1.396G |
| TangleNAS | AutoFormer-T | **78.842** | 98.249 | **88.290** | **98.066** | **92.347** | **92.396** | 8.98108M | 2.00G |
| SPOS+ES | AutoFormer-S | 81.700 | 99.10 | **90.459** | 97.90 | 94.8529 | **92.5447** | 22.9M | 5.1G |
| TangleNAS | AutoFormer-S | **81.964** | **99.12** | **90.459** | **98.3257** | **95.07** | 92.3707 | 28.806M | 6.019G |

Table 6: Evaluation on the AutoFormer-T space on downstream tasks. ImageNet accuracies are obtained through inheritance, whereas accuracies for the other datasets are achieved through fine-tuning the imagenet-pretrained model.

Table 5 and Table 6 evaluates TangleNAS against AutoFormer on the AutoFormer-T space. Interestingly, we observe that although AutoFormer sometimes outperforms TangleNAS upon inheritance from the supernet, the TangleNAS architectures are always better upon fine-tuning and much better upon retraining. For ImageNet-1k we obtain an accuracy of 78.842% by inheriting the weights of the optimal architecture in contrast to 75.474% by unconstrained evolutionary search on the on AutoFormer-T space (Chen et al., 2021a), i.e. a net improvement of 3.368% (see Table 6).

### 4.3.2 MOBILENETV3

Next, we study a convolutional search space based on the MobileNetV3 architecture. The search space is defined in Table 19 in the appendix and contains about $2 \times 10^{19}$ architectures. This follows from the search space designed by OFA (Cai et al., 2020), which searches for kernel-size, number of blocks and channel-expansion factor. Unlike OFA, we do not use the progressive shrinking scheme for the supernet training but activate all choices in our supernet at all times.

| Search Type | Optimizer | Top-1 acc | Params | FLOPS |
|---|---|---|---|---|
| Single-Stage | TangleNAS | **77.424** | 7.58M | 528.80M |
| Two-Stage | OFA+RS | 77.046 | 6.87M | 369.16M |
| | OFA+ES | 77.050 | 7.21M | 420.50M |
| Largest Arch | - | 77.336 | 7.66M | 566.17M |

Table 7: Evaluation on MobileNetV3

Table 7 shows that on this OFA search space, TangleNAS still outperforms OFA (based on both unconstrained evolutionary and random search on the pre-trained OFA supernet). Notably, TangleNAS even yields an architecture with higher accuracy than the largest architecture in the space (while OFA yields worse architectures).

### 4.3.3 LANGUAGE MODELLING SPACE

Finally, given the growing interest in efficiently training large language models and recent developments in scaling laws (Kaplan et al., 2020; Hoffmann et al., 2022), we study our efficient and scalable TangleNAS method on a language-model space.

We create our language model space around a smaller version of Andrej Karpathy's nanoGPT model[1]. In this transformer search space, we search for the embedding dimension, number of heads, number of layers and MLP ratios as defined in Table 18 (in the appendix). For each of these transformer search dimensions we consider 3 choices. We evaluate our small language model on the TinyStories (Eldan and Li, 2023) dataset. We improve over the SPOS+ES and SPOS+RS baselines as presented in Table 8. This improvement is statistically very significant with a two-tailed p-value of 0.0064 for ours v/s SPOS+ES and p-value of 0.0127 for ours v/s SPOS+RS.

## 5 RESULTS AND DISCUSSION

For a more in-depth evaluation, we now compare different properties of single-stage and two-stage methods, starting from their any-time performance, impact of the portion of train-val split on their performance and centered-kernel alignment of the supernet feature maps. Moreover, we also study the effect of pretraining, fine-tuning and retraining on the AutoFormer space for CIFAR10 and CIFAR100 and downstream performance of imagenet pre-trained best model on several classification datasets.

---

[1]https://github.com/karpathy/nanoGPT

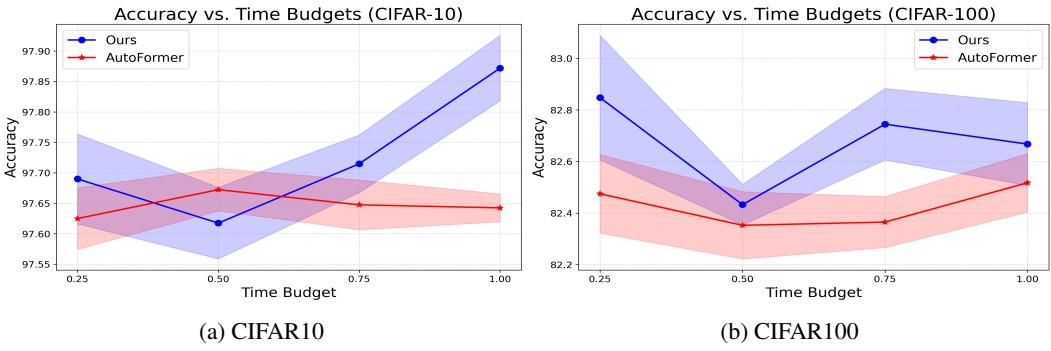

(a) CIFAR10      (b) CIFAR100

Figure 4: Any Time performance curves of AutoFormer vs. Ours

Finally, we discuss the insights derived from our single-stage approach in designing architectures on real world tasks.

**Space and time complexity** In practice, we observe that vanilla gradient-based NAS methods are memory and compute expensive in comparison to both two-stage methods and our TangleNAS approach with weight-superposition (described in Figure 2). In particular on the NB201 and DARTS search spaces we observe a 25.276% and 35.545% reduction in memory requirements for TangleNAS over DrNAS with WS. This issue only exacerbates for weight-entangled spaces like AutoFormer and MobileNetV3, making the application of vanilla gradient-based methods prac-

| Search Type | Optimizer | Test loss | Perplexity ↓ | Params | Inference Time |
|---|---|---|---|---|---|
| Single-Stage | TangleNAS | 1.412 ± 0.011 | 4.104 | 87.01M | 93.88s |
| Two-Stage | SPOS+RS | 1.433 ± 0.005 | 4.191 | 77.57M | 85.169s |
| | SPOS+ES | 1.444 ± 0.013 | 4.238 | 78.75M | 87.10s |

Table 8: Comparison of single and two stage methods on language model search space. We report the test loss and perplexity on the tinystories dataset

tically infeasible. This is because vanilla gradient-based methods incur an $\mathcal{O}(n)$ overhead over TangleNAS in terms of space and time complexity, where n is the number of operation choices.

**Any-Time performance** An essential characteristic of any neural architecture search (NAS) method is the attainment of strong *any-time performance*. End users of NAS techniques frequently prioritize the expedited identification of competitive architectures within limited time constraints. This becomes particularly crucial in light of the escalating computational expenses associated with training sophisticated and large neural networks, such as Transformers. Strong any-time performance is a pivotal criterion for the effective applicability of NAS in such resource-intensive domains. We thus compare the any-time performance of TangleNAS. Figure 3 demonstrates that TangleNAS (DrNAS+WE) is faster than its baseline method DrNAS+WS. Similarly, Figure 4 shows that TangleNAS has better anytime performance than AutoFormer.

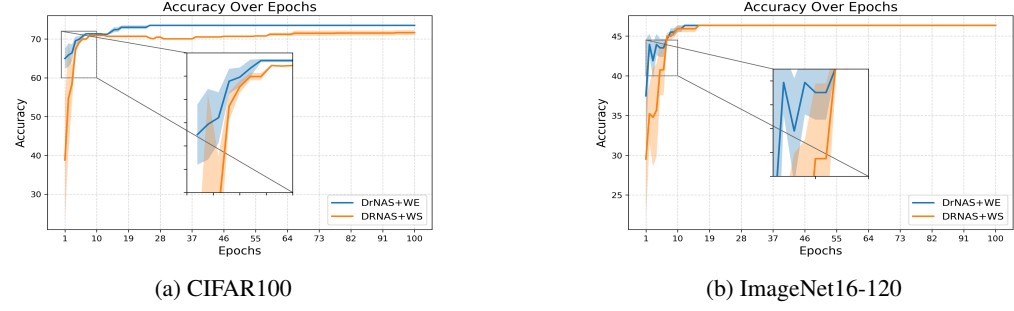

(a) CIFAR100      (b) ImageNet16-120

Figure 3: Test accuracy evolution over epochs for NB201

**Effect of fraction of training data** One-shot NAS commonly employs a 50%-50% train-valid split for cell-based spaces and 80%-20% for weight-entangled spaces. To avoid potential confounding

factors, we also study our method across different data splits for each search space. The results reported in Appendix A are largely similar across splits. In particular, we observe that single-stage methods are quite robust and performant across training fractions and search spaces, in comparison to two-stage methods.

**Architecture design insights** In transformer spaces reducing the *mlp-ratio* in the initial layers has a relatively low impact on performance (Figure 5) and can often work competitively or outperform handcrafted architectures. This observation is consistent across ViT and Language Model spaces. Conversely, *number of heads* and *embedding dimension* have a significant impact on the quantitative metric of choice. Pruning a few of the final layers also has a relatively low impact on performance. In the MobileNetV3 space, we find a strong preference for larger *number of channels* and larger *network depth*. In contrast, we discover that scaling laws for transformers may not necessary apply in convolutional spaces especially for *kernel sizes*. We observe that earlier layers favor 5x5 kernel sizes while later ones prefer 7x7 kernel sizes (3 being the smallest and 7 the largest).

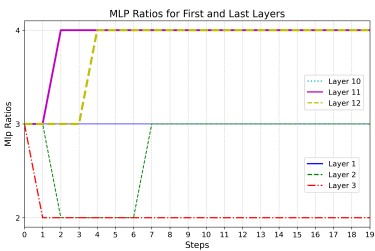

Figure 5: MLP ratio trajectory for LM. Number of layers range from 1-12 and mlp ratio choice can be 2,3 or 4.

**Effect of pretraining, fine-tuning and retraining** We study the impact of inheriting, fine-tuning, and retraining on the CIFAR10 and CIFAR100 datasets on the AutoFormer space. We observe that retraining almost always outperforms fine-tuning and inheriting. This brings to question the rank correlation between the inherited accuracy and retraining accuracy of two-stage methods and the training interference (Xu et al., 2022) in two-stage methods. Strong rank-correlation is desirable especially in two-stage methods as these methods rely on the performance proxy upon inheritance for the black-box search. Furthermore, we observe that while the SPOS+RS and SPOS+ES methods are performant upon inheritance, TangleNAS outperforms these models after fine-tuning and training from scratch. Improved supernet training properties for single-stage methods is further supported by Centered Kernel Alignment (CKA) analysis (Kornblith et al., 2019) (Table 25).

**ImageNet-1k pre-trained architecture on downstream tasks** Finally, we study the impact on fine-tuning the best model obtained from the search on downstream datasets. We follow the fine-tuning pipeline proposed in AutoFormer and fine-tune on different fine and coarse-grained datasets. We observe from Table 6 that the architecture discovered by TangleNAS on ImageNet is much more performant in fine-tuning to various datasets (CIFAR-10, CIFAR-100, Flowers, Pets and Cars) than the architecture discovered by SPOS.

## 6 CONCLUSION

In this paper, we investigate two of the most widely used NAS methods: *single-stage* and *two-stage*, each traditionally applied to different types of search spaces. We extend the applicability of single-stage NAS methods to weight-entangled spaces, a domain primarily explored by two-stage methods. By doing so, we aim to bridge the gap between these two distinct sub-fields. We empirically evaluate our proposed single-stage method, TangleNAS, on a diverse set of weight-entangled search spaces and tasks, showcasing its ability to outperform conventional two-stage NAS methods while enhancing search efficiency. Furthermore, we highlight the advantages of introducing weight-entanglement into traditional cell-based spaces, whenever feasible. Doing so not only improves the speed of single-stage methods but also reduces memory consumption. The modular nature of TangleNAS enables it to accommodate any differentiable optimizer and hence take advantage of the continually-improving set of methods for gradient-based NAS. Our results on various macro-level search spaces are already very encouraging and we therefore hope that this line of work will enable the NAS community to further improve modern architectures like Transformers.

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
