## A TRAINING ACROSS DATA FRACTIONS

| Search Type | Optimizer | Train portion | CIFAR10 | CIFAR100 |
|---|---|---|---|---|
| Single-Stage | TangleNAS | 50% | $97.715 \pm 0.0877$ | $82.5375 \pm 0.11764$ |
| | | 80% | $\mathbf{97.872 \pm 0.05369}$ | $\mathbf{82.6675 \pm 0.16073}$ |
| Two-Stage | SPOS+RS | 50% | $97.6799 \pm 0.0262$ | $82.5372 \pm 0.27954$ |
| | | 80% | $97.7674 \pm 0.0239278$ | $82.210 \pm 0.14242$ |
| | SPOS+RE | 50% | $97.77 \pm 0.03752$ | $82.354 \pm 0.11974$ |
| | | 80% | $97.6425 \pm 0.02286$ | $82.5175 \pm 0.1135$ |

Table 9: Evaluation on the AutoFormer-T space for CIFAR10 and CIFAR100

| Search Type | Optimizer | Train portion | Supernet | Accuracy (%) | | |
|---|---|---|---|---|---|---|
| | | | | CIFAR10 | CIFAR100 | ImageNet16-120 |
| Single-Stage | DrNAS | 50% | WS | $\mathbf{94.36 \pm 0.00}$ | $72.245 \pm 0.7315$ | $\mathbf{46.37 \pm 0.00}$ |
| | | 80% | | $\mathbf{94.36 \pm 0.00}$ | $71.1525 \pm 0.697$ | $\mathbf{46.37 \pm 0.00}$ |
| | TangleNAS | 50% | WE | $\mathbf{94.36 \pm 0.00}$ | $\mathbf{73.51 \pm 0.000}$ | $\mathbf{46.37 \pm 0.00}$ |
| | | 80% | | $\mathbf{94.36 \pm 0.00}$ | $\mathbf{73.51 \pm 0.000}$ | $\mathbf{46.37 \pm 0.00}$ |
| Two-Stage | SPOS+RS | 50% | WE | $89.107 \pm 0.88392$ | $56.865 \pm 2.5968$ | $31.665 \pm 1.1460$ |
| | | 80% | | $87.778 \pm 2.4462$ | $53.68 \pm 4.1736$ | $30.5449 \pm 3.6425$ |
| | SPOS+ES | 50% | WE | $87.133 \pm 2.6050$ | $56.463 \pm 2.3417$ | $29.7849 \pm 3.0149$ |
| | | 80% | | $89.095 \pm 0.8250$ | $56.363 \pm 4.7244$ | $30.935 \pm 3.54648$ |

Table 10: Comparison of test accuracies of single and two stage methods with WS and WE on NB201 search space

| Search Type | Optimizer | Train portion | Supernet type | Test acc |
|---|---|---|---|---|
| Single-Stage | DrNAS | 50% | WS | $91.19 \pm 0.0490$ |
| | | 80% | | $91.125 \pm 0.0334$ |
| | TangleNAS | 50% | WE | $\mathbf{91.3 \pm 0.023}$ |
| | | 80% | | $91.065 \pm 0.1633$ |
| Two-Stage | SPOS+RS | 50% | WE | $90.68 \pm 0.25345$ |
| | | 80% | | $90.687 \pm 0.10969$ |
| | SPOS+ES | 50% | WE | $90.3175 \pm 0.2233$ |
| | | 80% | | $90.595 \pm 0.2188$ |
| Optimum | - | - | - | 91.63 |

Table 11: Evaluation on toy cell-based search space on fashion-mnist dataset

| Search Type | Optimizer | Train portion | Supernet type | Test acc |
|---|---|---|---|---|
| Single-Stage | DrNAS | 50% | WS | 10% |
| | | 80% | | 10% |
| | TangleNAS | 50% | WE | $\mathbf{83.020 \pm 0.000}$ |
| | | 80% | | $82.495 \pm .46074$ |
| Two-Stage | SPOS+RS | 50% | WE | $81.253 \pm 0.67174$ |
| | | 80% | | $81.345 \pm 0.38336$ |
| | SPOS+ES | 50% | WE | $81.890 \pm 0.8002$ |
| | | 80% | | $82.322 \pm 0.60356$ |
| Optimum | - | - | - | 84.41 |

Table 12: Evaluation on toy conv-macro search space on CIFAR10 dataset

| Search Type | Optimizer | Train portion | Supernet | CIFAR10 | ImageNet |
|---|---|---|---|---|---|
| Single-Stage | TangleNAS | 50% | WE | $\mathbf{2.556 \pm 0.03406}$ | $\mathbf{25.69}$ |
| | | 80% | | $2.67 \pm 0.0756$ | 25.742 |
| | DrNAS | 50% | WS | $2.625 \pm 0.0750$ | 26.29 |
| | | 80% | | $2.580 \pm 0.02798$ | $\mathbf{25.67}$ |
| Two-Stage | SPOS+RS | 50% | WE | $2.965 \pm 0.07193$ | 27.114 |
| | | 80% | | $2.965 \pm 0.07193$ | 27.114 |
| | SPOS+RE | 50% | WE | $3.20000 \pm 0.06557$ | 27.424 |
| | | 80% | | $3.00249 \pm 0.037052$ | 26.76 |

Table 13: Comparison of test errors of single and two stage methods with WS and WE on DARTS search space

# B    SEARCH AND TRAINING TIMES

## B.1    NB201 AND DARTS

| Search Type | Optimizer | Supernet | Search Time (hrs) |
|---|---|---|---|
| Single-Stage | DrNAS | WS | 20.9166 |
| | TangleNAS | WE | 20.3611 |
| Two-Stage | SPOS+RS | WE | 29.51388 |
| | SPOS+ES | WE | 26.74721 |

Table 14: Comparison of search times NB201

| Search Type | Optimizer | Supernet | GPU Mem% | Search Time(hrs) |
|---|---|---|---|---|
| Single-Stage | DrNAS | WS | 81.08% | 29.4444 |
| | TangleNAS | WE | 52.26% | 25.472 |
| Two-Stage | SPOS+RS | WE | 25.66% | 18.7444 |
| | SPOS+ES | WE | 25.66% | 14.794 |

Table 15: Comparison of test errors of single and two-stage methods on the DARTS search space

# C    SEARCH SPACE DETAILS

**Toy cell space**    This particular search space takes its inspiration from the prominently used Differentiable Architecture Search (DARTS) space, and is composed of diminutive triangular cells, with each edge offering four choices of operations: (a) Separable 3x3 Convolution, (b) Separable 5x5 Convolution, (c) Dilated 3x3 Convolution, and (d) Dilated 5x5 Convolution. The macro-architecture of the model comprises of three cells of types reduction, normal, and reduction again stacked together. Notably, we entangle the 3x3 and 5x5 kernel weights for each operation type, i.e., separable convolutions and dilated convolutions. We evaluate these search spaces and their architectures on the Fashion-MNIST dataset by creating a small benchmark which we release here.

**Toy conv-macro space**    This toy space draws its inspiration from MobileNet-like spaces where we search for the number of channels and the kernel size of convolutional layers in a network (also called macro search space) for four convolutional layers. Every convolutional layer has a choice of three kernel sizes and number of channels. See Table 16 for more details. We evaluate this search space on the CIFAR10 dataset by creating a small benchmark which we release here.

| Architectural Parameter | Choices |
|---|---|
| Kernel sizes (all layers) | 3, 5, 7 |
| Number of channels (layer 1) | 8, 16, 32 |
| Number of channels (layer 2) | 16, 32, 64 |
| Number of channels (layer 3) | 32, 64, 128 |
| Number of channels (layer 4) | 64, 128, 256 |

Table 16: Toy Convolutional-Macro Search Space

**NASBench201 Space**    The NASBench201 (Dong and Yang, 2020) has a single cell type which is stacked 5 times. Each cell has 4 nodes and each node are connected by operations from amongst skip connection, conv 3x3, conv 1x1, max-pool or average-pool. Opportunities for weight entanglement in this space is limited and we entangle the conv 3x3 and conv 1x1 kernel on every edge for every cell, thus obtaining parameter savings over traditional weight sharing.

**DARTS Search Space**  The DARTS (Liu et al., 2019) search space has a cell with 13 possible edges stacked 8 times. The nodes of the cell are connected by operations amongst :none, max_pool_3x3, avg_pool_3x3, skip_connect, sep_conv_5x5, sep_conv_3x3, dil_conv_5x5, dil_conv_3x3.

**AutoFormer Space and Language Model Space**  We present the details of the AutoFormer Space and the Language Model space is Table 17 and Table 18 respectively.

| Architectural Parameter | Choices |
|---|---|
| Embedding dimension | 192, 216, 240 |
| Number of layers | 12, 13, 14 |
| MLP ratio (per layer) | 3.5, 4 |
| Number of heads (per layer) | 3, 4 |

Table 17: Choices for AutoFormer-T Search Space

| Architectural Parameter | Choices |
|---|---|
| Embedding Dimension | 384, 576, 768 |
| Number of Heads | 6, 8, 12 |
| MLP Expansion Ratio | 2, 3, 4 |
| Number of Layers | 5,6,7 |

Table 18: Choices for Language Model Space

| Architectural Parameter | Choices |
|---|---|
| Kernel sizes (all layers) | 3, 5, 7 |
| Channel expansion (all layers) | 3,4,6 |
| Number of blocks | 2,3,4 |

Table 19: MobileNet Search Space

## D  DETAILS ON METHOD

Traditional cell-based search spaces primarily consider independent operations (e.g., convolution or skip). One-shot differentiable optimizers thus have their mixture operations tailored to these search spaces which are not general enough to be applied to macro-level architectural parameters. Consider, e.g., the task of searching for the embedding dimension and the expansion ratio for a transformer. Here, a single operation, i.e., a linear expansion layer, has two different architectural parameters - one corresponding to the choice of embedding dimension and the other to the expansion ratio. To adapt single-stage methods to these *combined* operation choices in the search space, we propose the *combi-superposition operation*.

The combi-superposition operation simply takes the cross product of architectural parameters for the embedding dimension and expansion ratio and assigns its elements to *every combination* of these dimensions, allowing us to optimize jointly in this space without the need for a separate forward pass for each combination. Every combination maps to a unique sub-matrix of the operator weight matrix, indexed using both the embedding dimension and the expansion ratio. To address shape mismatches of the different operation weights during forward passes, every sub-matrix is zero-padded to match the shape of the largest matrix. See Algorithm 2 for more details, and Figure 2(b) for an overview of the idea.

---

**Algorithm 2** Combi-Superposition Operation
___

$embed\_dim \leftarrow [e_1, e_2, e_3, \ldots, e_n]$ {Choices for embedding dimension}
$expansion\_ratio \leftarrow [r_1, r_2, r_3, \ldots, r_m]$ {Choices for expansion ratio}
$\alpha \leftarrow [\alpha_1, \alpha_2, \alpha_3, \ldots, \alpha_n]$ {Architecture parameters for embedding dimension}
$\beta \leftarrow [\beta_1, \beta_2, \beta_3, \ldots, \beta_m]$ {Architecture parameters for expansion ratio}
$X \leftarrow input\_feature$
$W, b \leftarrow fc\_layer\_weight, fc\_layer\_bias$
$W_{mix} \leftarrow \mathbf{0}$
$b_{mix} \leftarrow \mathbf{0}$
$\alpha, \beta \leftarrow \text{normalize}(\alpha), \text{normalize}(\beta)$ {Normalize architecture parameters (e.g., using softmax)}
$\alpha\beta \leftarrow \alpha \times \beta$ {Cross product of $\alpha$ and $\beta$ for dimension $n \times m$}
**for** $i \leftarrow 1$ to $n \times m$ **do**
    $W\_i = W[: (embed\_dim[i] \times expansion\_ratio[i]), : embed\_dim[i]]$
    $b\_i = b[: embed\_dim[i] \times expansion\_ratio[i]]$
    $W_{mix} = W_{mix} + \alpha\beta[i] \times \text{PAD}(W_i)$
    $b_{mix} = b_{mix} + \alpha\beta[i] \times \text{PAD}(b_i)$
**end for**
$Y \leftarrow X \cdot W_{mix} + b_{mix}$ {Compute the output of the FC layer with a mixture of weights and bias}
**return** $Y$

---

For completeness and for comparison with TangleNAS (Algorithm 1), we present Algorithm 3 which describes a generic two-stage method on a macro search space with WE. Also, vanilla single-stage methods on cell-based WS spaces follow Algorithm 4.

**Algorithm 3** Weight Entanglement (Two-Stage)

1: **Input:** $M \leftarrow$ number of cells, $N \leftarrow$ number of operations
$\quad\quad \mathcal{O} \leftarrow [o_1, o_2, o_3, ...o_N]$
$\quad\quad \mathcal{W}_{\max} \leftarrow \cup_{i-1}^N w_i$
$\quad\quad \eta =$ learning rate of $\mathcal{W}_{\max_{\mathcal{O}}}$
2: $Cell_j \leftarrow DAG(\mathcal{O}_|, \mathcal{W}_{\max_|})$ /* defined for j=1...M */
3: $Supernet \leftarrow \cup_i^M Cell_i$
4: /* example of forward propagation on the cell */
5: **for** $j \leftarrow 1$ to $M$ **do**
6: $\quad i \sim \mathcal{U}(1, N)$
7: $\quad$ /* Slice weight matrix corresponding to operation */
8: $\quad o_j(x, \mathcal{W}_{max}) = o_{(j,i)}(x, \mathcal{W}_{max}[: i])$
9: **end for**
10: /* weights update */
11: $\mathcal{W}_{\max}[: i] = \mathcal{W}_{\max}[: i] - \eta \nabla_{\mathcal{W}_{\max}}[: i]\mathcal{L}_{train}(\mathcal{W}_{\max})$
12: /* Search */
13: $Supernet^* \leftarrow$ pre-trained supernet
14: $selected\_arch \leftarrow$ Evolutionary-Search($Supernet^*$)

**Algorithm 4** Weight Sharing (Single-Stage)

1: **Input:** $M \leftarrow$ number of cells, $N \leftarrow$ number of operations
$\quad\quad \mathcal{O} \leftarrow [o_1, o_2, o_3, ...o_N]$
$\quad\quad \mathcal{W}_{\mathcal{O}} \leftarrow [w_1, w_2, w_3, ....w_N]$
$\quad\quad \mathcal{A} \leftarrow [\alpha_1, \alpha_2, \alpha_3, ...\alpha_N]$
$\quad\quad \gamma =$ learning rate of $\mathcal{A}$
$\quad\quad \eta =$ learning rate of $\mathcal{W}_{\mathcal{O}}$
$\quad\quad f$ is a function or distribution s.t. $\sum_{i=1}^N f(\alpha_i) = 1$
2: $Cell_i \leftarrow DAG(\mathcal{O}_|, \mathcal{W}_{\mathcal{O}_|})$ /* defined for i=1...M */
3: $Supernet \leftarrow \cup_i^M Cell_i \cup \mathcal{A}$
4: /* example of forward propagation on the cell */
5: **for** $j \leftarrow 1$ to $M$ **do**
6: $\quad$ /* Compute mixture operation as weighted sum of output of operations*/
7: $\quad \overline{o_j(x, \mathcal{W}_{\mathcal{O}})} = \sum_{i=1}^N f(\alpha_i) \, o_{(j,i)}(x, w_{(j,i)})$
8: **end for**
9: /* weights and architecture update */
10: $\mathcal{A} = \mathcal{A} - \gamma \nabla_{\mathcal{A}} \mathcal{L}_{val}(\mathcal{W}_{\mathcal{O}}^*, \mathcal{A})$
11: $\mathcal{W}_{\mathcal{O}} = \mathcal{W}_{\mathcal{O}} - \eta \nabla_{\mathcal{W}_{\mathcal{O}}} \mathcal{L}_{train}(\mathcal{W}_{\mathcal{O}}, \mathcal{A})$
12: /* Search */
13: $selected\_arch \leftarrow \arg\max(\mathcal{A})$

# E EXPERIMENTAL SETUP

## E.1 TOY SEARCH SPACES

| Search Space | Cell Space | Conv Macro |
|---|---|---|
| Epochs | 100 | 100 |
| Learning rate (LR) | 0.1 | 3e-4 |
| Min. LR | 0.001 | 0.0001 |
| Optimizer | SGD | Adam |
| Architecture LR | 0.0003 | 0.0003 |
| Batch Size | 64 | 64 |
| Momentum | 0.9 | 0.9 |
| Nesterov | True | False |
| Weight Decay | 0.0005 | 0.0005 |
| Arch Weight Decay | 0.001 | 0.001 |
| Regularization Type | L2 | L2 |
| Regularization Scale | 0.001 | 0.001 |

Table 20: Configurations used in the DrNAS experiments on Toy Spaces.

| Search Space | Cell Space | Conv Macro |
|---|---|---|
| Epochs | 250 | 250 |
| Learning rate (LR) | 0.1 | 3e-4 |
| Min. LR | 0.001 | 0.0001 |
| Optimizer | SGD | Adam |
| Batch Size | 64 | 64 |
| Momentum | 0.9 | 0.9 |
| Nesterov | True | False |
| Weight Decay | 0.0005 | 0.0005 |

Table 21: Configurations used in the SPOS experiments on Toy Spaces.

## E.2 LANGUAGE MODEL

We use the AdamW optimizer in all experiments pertaining to language modelling. Other hyperparameter choices are as in Table 22.

| Search Space | Small-LM |
|---|---|
| Learning rate (LR) | 5e-4 |
| Min LR | 5e-5 |
| Beta2 | 0.99 |
| Warmup Iters | 100 |
| Max Iters | 6000 |
| Lr decay iters | 6000 |
| Batch size | 12 |
| Weight decay | 1e-1 |

Table 22: Configurations used in DrNAS on the Language Model Spaces

## E.3 AUTOFORMER AND OFA

We use a 50%-50% train and validation split for the CIFAR10 and CIFAR100 dataset for the cell-based spaces and a 80%-20% for the weight-entangled spaces We use the official source code of AutoFormer available at code for all the AutoFormer experiments on CIFAR10 and CIFAR100. We closely follow the AutoFormer training pipeline and search space design. AutoFormer searched on three transformer sizes Autoformer-T (tiny), AutoFormer-S (small) and Autoformer-B (base). We currently restrict ourselves to Autoformer-T. For baselines like OFA and AutoFormer we follow their respective recipes to obtain the train-valid split for ImageNet-1k. Our models were trained on 2xA100s with the same effective batch-size as AutoFormer. For MobileNetV3 from once-for-all we use the same training hyperparameters as the baseline as here (in addition to architectural parameters same as Table 23) .

### E.3.1 AUTOFORMER FINE-TUNING

**C10 and C100 pretrained supernet** We fine-tune the C10 and C100 selected networks (after inheriting them from the supernet) for 500 epochs. We set the learning rate to 1e-3, the warmup epochs to 5 , the warmup learning rate to 1e-6 and the min learning rate to 1e-5. All other hyperparameters are set same as Appendix E.3.

**ImageNet pretrained supernet** We follow the DeiT (Touvron et al., 2021) finetuning pipeline as used in AutoFormer. Mainly we set the epochs to 1000, the warmup-epochs to 5 , the scheduler to cosine, the mixup to 0.8, the smoothing to 0.1, the weight-decay to 1e-4, the batch-size 64, the optimizer to sgd, the learning rate to 0.01 and the warmup-lr to 0.0001 for all datasets.

## E.4 NB201 AND DARTS

For single-stage optimizers, the supernet was trained with four different seeds. The supernet with the best validation performance from these four was discretized to obtain the final model, which is then trained from scratch four times to obtain the results shown in the table. For two-stage methods, yet again, we train the supernet four times, and perform Random Search (RS) and Evolutionary Search (ES) on each of them. The best model obtained across all four supernets for both methods are then trained from scratch with four seeds to compute the final results. For each DrNAS we follow the same recipe as suggested by the authors across all search spaces. To accommodate for multiple training recipes, we write a configurable training pipeline. The configurations for DrNAS, and SPOS are shown in Table 23 and Table 24, respectively.

| Search Space | DARTS | NB201 |
|---|---|---|
| Epochs | 50 | 100 |
| Learning rate (LR) | 0.1 | 0.025 |
| Min. LR | 0.0 | 0.001 |
| Architecture LR | 0.0006 | 0.0003 |
| Batch Size | 64 | 64 |
| Momentum | 0.9 | 0.9 |
| Nesterov | True | False |
| Weight Decay | 0.0003 | 0.0003 |
| Arch Weight Decay | 0.001 | 0.001 |
| Partial Connection Factor | 6 | - |
| Regularization Type | L2 | L2 |
| Regularization Scale | 0.001 | 0.001 |

Table 23: Configurations used in the DrNAS experiments.

| Search Space | DARTS | NB201 |
|---|---|---|
| Epochs | 250 | 250 |
| Learning rate (LR) | 0.025 | 0.025 |
| Min. LR | 0.001 | 0.001 |
| Architecture LR | 0.0003 | 0.0003 |
| Batch Size | 256 | 64 |
| Momentum | 0.9 | 0.9 |
| Nesterov | True | True |
| Weight Decay | 0.0005 | 0.0005 |
| Arch Weight Decay | 0.001 | 0.001 |

Table 24: Configurations used in the SPOS experiments.

## F  OPTIMAL ARCHITECTURES DERIVED

### F.1  DARTS

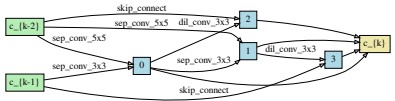

Figure 6: DRNAS with WE normal cell

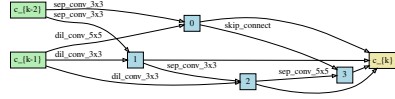

Figure 7: DRNAS with WE reduction cell

### F.2  SMALL LM

num_layers: 7, embed_dim: 768, num_heads: [12, 12, 12, 12, 12, 12, 12], mlp_ratio: [3, 4, 4, 4, 4, 4, 4]

### F.3 AUTOFORMER

#### F.3.1 CIFAR10

**50%50% split** mlp_ratio:[4, 4, 4, 4, 4, 4, 4, 4, 4, 3.5, 4, 4, 3.5, 4], num_heads:[4, 4, 4, 4, 4, 4, 4, 4, 4, 4, 4, 4, 4, 4], num_layers: 14 embed_dim: 216

**80%-20% split** mlp_ratio:[4, 4, 4, 4, 4, 4, 4, 4, 3.5, 4, 4, 4, 3.5, 3.5], num_heads:[4, 4, 4, 4, 4, 4, 4, 4, 4, 4, 4, 4, 4, 4], depth: 14, embed_dim: 240

#### F.3.2 CIFAR100

**50%-50% split** mlp_ratio: [4,4,4,4,4,4,4,4,3.5,4,4,4,4,4], num_heads:[4,4,4,4,4,4,4,4,4,4,4,4,4,4], depth: 14, embed_dim: 216

**80%-20% split** mlp_ratio:[3.5,4,4,4,4,4,4,4,4,4,4,4,4,4], num_heads:[4,4,4,4,4,4,4,4,4,4,4,4,4,4, depth: 14, embed_dim: 240

#### F.3.3 IMAGENET1-K

mlp_ratio:[4,4,4,4,4,4,4,4,4,4,4,4,4,4], num_heads:[4,4,4,4,4,4,4,4,4,4,4,4,4,4, depth: 14, embed_dim: 240

### F.4 MOBILENETV3

Kernel_sizes:[7,5,5,7,5,5,7,7,5,7,7,7,5,7,7,5,5,7,7,5],Channel_expansion_factor:[6,6,6,6,6,6,6,6,6,6,6,6,6,6,6,6,6,6,6,6], Depths : [4, 4, 4, 4, 4]

### F.5 TOY SPACES

#### F.5.1 TOY CELL (OUR BEST ARCHITECTURE)

**50%-50% split** Genotype(normal=[('dil_conv_3x3', 0), ('dil_conv_3x3', 0), ('sep_conv_3x3', 1)], normal_concat=range(1, 3), reduce=[('sep_conv_3x3', 0), ('sep_conv_3x3', 0), ('dil_conv_3x3', 1)], reduce_concat=range(1, 3))

**80%-20% split** Genotype(normal=[('dil_conv_3x3', 0), ('dil_conv_5x5', 0), ('sep_conv_3x3', 1)], normal_concat=range(1, 3), reduce=[('sep_conv_3x3', 0), ('sep_conv_5x5', 0), ('dil_conv_3x3', 1)], reduce_concat=range(1, 3))

### F.6 TOY CONV-MACRO (OUR BEST ARCHITECTURE)

**50%-50% split:** Channels = [32, 64, 128, 64], Kernel Sizes = [5, 5, 7, 7].

**Train-Val fraction 80%-20%:** Channels = [32, 64, 128, 64], Kernel Sizes = [5, 5, 7, 7].

## G ARCHITECTURE REPRESENTATION ANALYSIS

**CKA** Centered Kernel Alignment (CKA) (Kornblith et al., 2019) is a metric, based on the Hilbert- Schmidt Independence Criterion (HSIC). This metric is primarily designed to model similarity between representations in neural networks. In this section we study the CKA between structurally identical layers in the inherited, fine-tuned and retrained networks in the AutoFormer-T space. Precisely, the goal here is to study how similar the inherited and fine-tuned representations are to the actual model trained from scratch. Table 25 Presents the average CKA values for a fixed subsample of the CIFAR10 and CIFAR100 dataset. We observe that the representations learned by the single-stage supernet are more similar to the architecture finetuned or trained from scratch. This further emphasizes the potential issues with using the inherited weights from the supernet as proxy for search for two-stage methods as highlighted in (Xu et al., 2022).

| Model | Inherit v/s Retrain | | Fine-Tune v/s Retrain | |
|---|---|---|---|---|
| | CIFAR10 | CIFAR100 | CIFAR10 | CIFAR100 |
| TangleNAS | **0.4630** | **0.5853** | 0.57125 | **0.65275** |
| SPOS+ES | 0.45812 | 0.57932 | 0.569374 | 0.6309 |
| SPOS+RS | 0.44124 | 0.583 | **0.5797** | 0.638948 |

Table 25: CKA correlation between layers

## H    NAS BEST PRACTICES CHECKLIST

We now describe how we addressed the individual points of the NAS best practice checklist (Lindauer and Hutter, 2020).

1. **Best Practices for Releasing Code**

   For all experiments you report:
   (a) Did you release code for the training pipeline used to evaluate the final architectures? Yes
   (b) Did you release code for the search space? Yes
   (c) Did you release the hyperparameters used for the final evaluation pipeline, as well as random seeds? Yes
   (d) Did you release code for your NAS method? Yes
   (e) Did you release hyperparameters for your NAS method, as well as random seeds? Yes

2. **Best practices for comparing NAS methods**

   (a) For all NAS methods you compare, did you use exactly the same NAS benchmark, including the same dataset (with the same training-test split), search space and code for training the architectures and hyperparameters for that code? Yes
   (b) Did you control for confounding factors (different hardware, versions of DL libraries, different runtimes for the different methods)? Yes
   (c) Did you run ablation studies? Yes
   (d) Did you use the same evaluation protocol for the methods being compared? Yes
   (e) Did you compare performance over time? Yes
   (f) Did you compare to random search? Yes. Black box random search is very expensive in our case, hence we use the supernet from single-path-one-shot(SPOS) for random search.
   (g) Did you perform multiple runs of your experiments and report seeds? Yes
   (h) Did you use tabular or surrogate benchmarks for in-depth evaluations? Yes

3. **Best practices for reporting important details**

   (a) Did you report how you tuned hyperparameters, and what time and resources this required? N/A. We did not tune any hyperparameters.
   (b) Did you report the time for the entire end-to-end NAS method (rather than, e.g., only for the search phase)? Yes
   (c) Did you report all the details of your experimental setup? Yes