# OpenReview forum: "Weight-Entanglement Meets Gradient-Based Neural Architecture Search"
_ICLR.cc/2024/Conference — Submitted to ICLR 2024_

### Official Review · Reviewer_JeZf · 2023-10-31

**Soundness:** 2 fair
**Presentation:** 2 fair
**Contribution:** 2 fair
**Rating:** 3
**Confidence:** 5

**Summary:**

This paper proposes the application of single-stage NAS methods to the weight-entanglement search space. The authors observe that weight-entanglement spaces are typically explored using two-stage methods, while cell-based spaces are usually explored using single-stage methods. The authors bridge the gap between them and conduct extensive experiments.

**Strengths:**

- The paper demonstrates the feasibility of the proposed method through experiments on different datasets and search spaces.
- The experimental results reveal interesting phenomena and observations.

**Weaknesses:**

- The motivation behind this work is not clear, and it appears to be a simple combination of existing methods, lacking innovation. What is the necessity and advantage of using single-stage search?
- The weight entanglement has a higher weight-sharing extent. Does the intensification of weight entanglement during the single-stage search process affect search performance?
- The description of the method is too simplistic, resulting in a lack of overall contribution.
- In the experimental section, the comparison with related works is not comprehensive enough, as it does not include some comparisons with methods based on weight entanglement and single-stage NAS.

**Questions:**

- What is the specific method referred to in Figure 2b? The description is unclear.
- What do LHS and RHS represent in Figure 2?

---

> ### Author Response · Authors · 2023-11-19
>
> We thank the reviewer for the detailed comments and suggestions. We respond to each of your questions below:
>
> > The motivation behind this work is not clear, and it appears to be a simple combination of existing methods, lacking innovation. What is the necessity and advantage of using single-stage search?
>
> We thank the reviewer for their question. To the best of our knowledge, our work is the first to efficiently and effectively apply single-stage methods to weight-entanglement/foundation-model macro spaces. This makes our approach novel and very useful in practice.
>
> Our work is primarily motivated by two goals:
>
> Firstly, we aim to make single-stage NAS methods amenable to foundation model macro spaces (language models, vision transformers, mobilenets). Note that single stage methods, even recent ones like [Lambda-DARTS](https://openreview.net/forum?id=oztkQizr3kk), have only been evaluated on cell-based spaces which are not very realistic in deep learning deployment scenarios. In our opinion, it is crucial to make single-stage methods more accessible to foundation models to ensure that advancement in these methods do not remain restricted to cell-based spaces. Further, our plug-and-play framework and our publicly available code makes integration evaluation of any single-stage optimizer on these spaces easy.
>
> Secondly, two-stage methods rely on performance proxies from the supernet which might not be highly correlated with the ground truth accuracies in practice. These supernets also face interference [1] during training which exacerbate these issues. We also conduct an analysis on the CKA[2] correlation between feature representations of the best architecture in our single-stage supernet and our two-stage supernet and find that our supernet in general shows better correlations.
>
> | Model        | Inherit v/s Retrain (CIFAR10) | Inherit v/s Retrain (CIFAR100) | Fine-Tune v/s Retrain (CIFAR10) | Fine-Tune v/s Retrain (CIFAR100) |
> |--------------|--------------------------------|---------------------------------|----------------------------------|-----------------------------------|
> | TangleNAS    | **0.4630**                     | **0.5853**                      | 0.57125                          | **0.65275**                       |
> | SPOS+ES      | 0.45812                        | 0.57932                         | 0.569374                         | 0.6309                            |
> | SPOS+RS      | 0.44124                        | 0.583                           | **0.5797**                       | 0.638948                          |
>
>
>
>
> [1]  Shipard, J., Wiliem, A. and Fookes, C., 2022. Does Interference Exist When Training a Once-For-All Network?. In Proceedings of the IEEE/CVF Conference on Computer Vision and Pattern Recognition (pp. 3619-3628).
>
> [2] Kornblith, S., Norouzi, M., Lee, H. and Hinton, G., 2019, May. Similarity of neural network representations revisited. In International conference on machine learning (pp. 3519-3529). PMLR.

---

> > ### Author Response · Authors · 2023-11-19
> >
> > >The weight entanglement has a higher weight-sharing extent. Does the intensification of weight entanglement during the single-stage search process affect search performance?
> >
> > In cell-based spaces and toy spaces intensification of weight entanglement improves performance as shown in Tables 1, 2, 3, and 4 . Interestingly, in NAS-Bench 201 (Table 3, Figure 3) we observe that intensification of weight entanglement using our approach leads to more robust convergence to the optimal in this space.
> >
> > > The description of the method is too simplistic, resulting in a lack of overall contribution.
> >
> > While we acknowledge that the method is simple, we think that this is the primary strength of our work. The novelty of the work stems from its ability to bridge research in end-to-end architectural parameter learning, like in DARTS, and supernet-based NAS methods. Our plug-and-play framework and our publicly available code makes integration and evaluation of any single-stage optimizer on these spaces easy. The simplicity of our work speaks for its applicability in terms of scaling single-stage methods now to more realistic foundation model macro architecture spaces.
> >
> > > In the experimental section, the comparison with related works is not comprehensive enough, as it does not include some comparisons with methods based on weight entanglement and single-stage NAS.
> >
> > We thank the reviewer for this comment. For every search space we compare with the respective 2-stage baseline i.e. AutoFormer and Once-For-All. To the best of our knowledge these are the primary baselines that single-stage approaches should match up to. We show very competitive performance in comparison to these baselines on a variety of different search spaces. We request the reviewer to please provide additional information/comments on the other possible weight-entanglement baselines on the above spaces.
> >
> > To the best of our knowledge, we are the first approach to scale single-stage methods effectively to ViTs, LLMs and MobileNet. Given our initial studies on competitive performance of DrNAS on a variety of spaces we restrict to DrNAS as the single-stage method of choice. Our method can exploit developments in single-stage NAS as better algorithms are developed due to the nature of our plug-and-play framework.
> >
> > If the reviewer has any further questions we would be happy to address them.

---

### Official Review · Reviewer_2nyZ · 2023-10-31

**Soundness:** 3 good
**Presentation:** 3 good
**Contribution:** 2 fair
**Rating:** 5
**Confidence:** 2

**Summary:**

The paper presents a gradient-based neural architecture search approach (TangleNAS) with a weight-entangled search space.  The main idea is to combine the memory efficiency of weight sharing (entanglement) with the search time efficiency of a differentiable (gradient-based) search space.  DrNAS, a gradient-based approach, is taken as a reference and extended to weight sharing by modifying the edge operations. All edge operations are summed after being individually weighted. The approach is evaluated on several standard benchmarks, where it often shows an improvement when combining both types of search spaces.

**Strengths:**

+ The paper is well written. In particular, the related work is complete and the proposed approach is clearly positioned in relation to existing approaches. In addition, the method is mostly well presented and the main idea is easy to follow.

+ The proposed idea of edge operations works well in practice. Moreover, the idea could be applied to various gradient-based NAS methods.

+ The experimental section considers several standard benchmarks for evaluation, both cell-based and macro search spaces. It shows an improvement in results compared to DrNAS for most cases.

**Weaknesses:**

- Contribution: As stated in the paper, the proposed approach is applicable to different gradient-based NAS methods. It would add value to the paper to demonstrate this. At the moment it looks like an approach to extend DrNAS, but at least DARTS and/or GDAS (or more recent TE-NAS) should have been considered.

-  On the macro search space, it would make sense to consider a macro approach and then introduce the cell-based part of the proposed method. It's not clear what the reference is for measuring the improvement. Nevertheless, the comparison with existing methods is useful.

- (Major limitation) This is a benchmark driven approach. It would therefore be useful to include the latest results on NAS, e.g. from Lukasik, Jovita, Steffen Jung and Margret Keuper. "Learning where to look - generative NAS is surprisingly efficient." European Computer Vision Conference. Cham: Springer Nature Switzerland, 2022. Then it would also be helpful to discuss why and when the paper lacks performance compared to the latest state-of-the-art approaches. For example, on the DARTS search space (Table 4), the current SOTA is much lower than the paper's results (the cited paper or TE-NAS, for example, perform better). It is therefore important to compare with the latest approaches and possibly improve on their setup.

- Clarity: The method re-defines the parts of the search space. For example, the superset search space and edge definition are missing. In general, the method would benefit from a section defining this problem.

**Questions:**

- It would be useful to know why the latest SOTA approaches have not been used as a reference for improvement using the proposed approach.

---

> ### Author Response · Authors · 2023-11-19
>
> We thank the reviewer for the detailed comments and suggestions. We respond to each of your questions below:
>
> > It would add value to the paper to demonstrate DARTS and/or GDAS…
>
> In the single-stage neural architecture search literature [1]  it is well-known that several of the single-stage optimizers like DARTS, GDAS etc.have many failure modes. We started our study on weight entanglement spaces by studying DrNAS, DARTS and GDAS on the two tiny search spaces which we create benchmarks for. We find from initial analysis that DrNAS performs competitively and is robust on both of these spaces in comparison to DARTS and GDAS. Hence we restrict ourselves to DrNAS in our large scale experiments. We have also evaluated DARTS and GDAS on the AutoFormer-T space on CIFAR-100 and presented the results below.
>
> Toy conv macro (CIFAR10)
> |  Optimizer | Test-acc          |
> |------------|-------------------|
> | DrNAS      | **83.02**         |
> | SPOS+RS    | 81.2525           |
> | SPOS+RE    | 81.89             |
> | DARTS_v1   | 81.61             |
> | DARTS_v2   | 81.49             |
> | GDAS       |  10% (degenerate) |
>
> Toy cell entangled (Fashion MNIST)
>
> |  Optimizer | Test-acc  |
> |------------|-----------|
> | DrNAS      |  **90.93** |
> | DARTS-V1   | 89.905    |
> | DARTS-V2   | 90.7475   |
> | GDAS       | 90.618    |
> | SPOS+RS   | 90.6875   |
> | SPOS+RE    | 90.595    |
>
> AutoFormer-T (CIFAR100)
> |  Optimizer | CIFAR100 Test-acc  |
> |------------|---------------|
> |TangleNAS-DARTS    |    82.107 ± 0.392        |
> | TangleNAS-GDAS    |     82.12 ± 0.2813        |
> | TangleNAS-DrNAS   |   **82.668 ± 0.161**        |
> | SPOS+ES   |       82.5175 ± 0.114      |
> | SPOS+RS  |    82.210 ± 0.14242          |
>
> Since our focus is mainly 2-stage gradient based NAS without the use of zero-cost proxies we restrict ourselves to approaches which operate in this realm (unlike TE-NAS, which uses proxies for pruning).
>
> > On the macro search space, it would make sense to consider a macro approach and then introduce the cell-based part of the proposed method. It's not clear what the reference is for measuring the improvement. Nevertheless, the comparison with existing methods is useful.
>
> We would like to clarify that none of the macro-spaces in our experimental setup (i.e. AutoFormer, MobileNet, LLM space) have any cell-based components. We operate on the same search spaces as AutoFormer or OFA only modifying the NAS optimizer itself with our proposed optimizer. The baselines we use are the two-stage counterparts (usually methods which are based on SPOS with some modifications). For completeness we compare against the AutoFormer’s original code, the Once-For-All’s original code as baselines when comparing their two-stage method with ours. For search spaces which do not have two-stage methods available we implement the Single-Path-One-Shot (SPOS) two-stage method in these spaces.
>
> > This is a benchmark driven approach …
>
> We compare against DrNAS which improves just with the introduction of weight-entanglement. We don’t compare with other baselines simply because the goal of our work is to show the effective applicability of single-stage methods on weight-entanglement spaces (transformers, mobilenet)  instead of achieving SOTA on cell-based spaces (DARTS, NB201). We hope that our flexible plug-and-play framework makes evaluation on new single-stage methods on more realistic and practical spaces like transformers, mobilenet easier.
>
> > Clarity
>
> We thank the reviewer for this suggestion. We will provide more details about the method in our updated manuscript.
>
> If the reviewer has any further questions we would be happy to address them.

---

> ### Author Response · Authors · 2023-11-21
>
> > Evaluation on other optimizers..
>
> We have now evaluated DARTS and GDAS on the AutoFormer-T space for CIFAR10 too. Similar to the results on CIFAR100, here too TangleNAS-DrNAS performs best out of the three choices of optimizers. This motivates our choice of gradient-based NAS optimizer in our experiments.
>
> AutoFormer-T (CIFAR10)
>
> |  Optimizer | CIFAR10 Test-acc  |
> |------------|---------------|
> |TangleNAS-DARTS    |    97.672 ± 0.04    |
> | TangleNAS-GDAS    |    97.45 ± 0.096       |
> | TangleNAS-DrNAS   |   **97.872 ± 0.054**        |
> | SPOS+ES   |       97.6425 ± 0.023      |
> | SPOS+RS  |    97.767 ± 0.024         |

---

### Official Review · Reviewer_b7V3 · 2023-11-01

**Soundness:** 3 good
**Presentation:** 3 good
**Contribution:** 3 good
**Rating:** 6
**Confidence:** 5

**Summary:**

In this paper, the authors introduced architectural parameters to supernet search space, where all operation choices are superposed ontothe largest one with weights. This simple process enables searching for optimal sub-network via gradient-based optimization. The authors applied the proposed method to MobileNetV3 and ViT search space and got promising performance.

**Strengths:**

The proposed method is simple yet effective. It inherits the good merits of DARTS-like methods (end-to-end learning with the help of architectural parameters) and supernet methods (memory efficient and supporting fine-grained search spaces).

**Weaknesses:**

Some experimental details are not clear:

- It is unclear from the paper what are the FLOPs and parameter sizes of the searched results in the experiments. Are the comparison are done between networks?

- Any reason why the performance of AutoFormer variants from that paper is not listed in Table 6? Again, there should be the FLOPs and parameter size of each model.

- On Table 7, there should be comparisons with some other works which use MobileNetV3 search space (e.g., AtomNAS), together with their FLOPs and parameter sizes. Why are the results of OFA with progressive shrink not used in the table?

**Questions:**

My main concern is the lack of some information and comparisons in the experiments, as mentioned above.

---

> ### Author Response · Authors · 2023-11-19
>
> We thank the reviewer for the detailed comments and suggestions. We respond to each of your questions below:
>
> > It is unclear from the paper what are the FLOPs and parameter sizes of the searched results in the experiments. Are the comparisons done between networks?
>
> We have now updated the manuscript to reflect these numbers (Table 5,6,7,8). In our experimental setup, we perform unconstrained search using both two-stage and single-stage methods (i.e. TangleNAS) and then compare their evaluations. The comparisons are done between individual networks in the supernet.
>
> Furthermore now we also conduct an experiment to search for models with smaller parameter sizes. We add a differentiable parameter penalty to the loss function to enforce this. We then perform the evolutionary search of SPOS setting the parameter size of the tanglenas model as constraint. We find out that the architecture SPOS discovers in this constrained version is still worse than the tanglenas architecture discovered. This shows the ability of our search method to discover better architectures even at smaller parameter budgets in its constrained version.
> | Optimizer | CIFAR10           | Params    | CIFAR100         | Params  |
> |-----------|-------------------|-----------|------------------|---------|
> | TangleNAS | 97.254925,0.11523 | 6.647626M | 81.25467±0.27151 | 7.08394 |
> | SPOS+RE   | 97.16425,0.13604  | 6.56041M   | 80.8324±0.23757  | 6.8866  |
>
> > Any reason why the performance of AutoFormer variants from that paper is not listed in Table 6? Again, there should be the FLOPs and parameter size of each model.
>
> We primarily restrict our evaluation to AutoFormer-T due to compute restrictions. However we have now evaluated our method on the AutoFormer-S space and the results (with resolution 224 images) is in Table-6 of the updated manuscript. Our approach again outperforms the architecture derived by the corresponding two-stage method. We present the table-6 below:
> | NAS Method | SuperNet-Type | ImageNet | CIFAR10 | CIFAR100 | Flowers | Pets | Cars | Params | FLOPS |
> |------------|---------------|----------|---------|----------|---------|------|------|--------|-------|
> | SPOS+ES    | AutoFormer-T  | 75.474   | 98.019  | 86.369   | **98.066** | 91.558 | 91.935 | 5.893M | 1.396G|
> | TangleNAS  | AutoFormer-T  | **78.842** | 98.249 | **88.290** | **98.066** | **92.347** | **92.396** | 8.98108M | 2.00G|
> | SPOS+ES    | AutoFormer-S  | 81.700   | 99.10   | **90.459** | 97.90  | 94.8529 | **92.5447** | 22.9M  | 5.1G  |
> | TangleNAS  | AutoFormer-S  | **81.964** | **99.12** | **90.459** | **98.3257** | **95.07** | 92.3707 | 28.806M | 6.019G|
>
>
>
> > On Table 7, there should be comparisons with some other works which use MobileNetV3 search space (e.g., AtomNAS), together with their FLOPs and parameter sizes. Why are the results of OFA with progressive shrink not used in the table?
>
> We do use the OFA-PS supernet and report the accuracy after evolutionary search using their code. The search space variant we use is the smaller MobileNetV3 space, i.e. ofa_mbv3_d234_e346_k357_w1.0 [here](https://github.com/mit-han-lab/once-for-all/) . To the best of our knowledge AtomNAS has a very different search space compared to OFA, hence we think that AtomNAS is not really comparable to OFA or our proposed approach.
>
> If the reviewer has any further questions we would be happy to address them.

---

> > ### Comment · Reviewer_b7V3 · 2023-11-22
> >
> > Thanks for providing the additional results. I have no more questions except a small one: in the updated table 6, it would be better if you can provide TangleNAS with similar parameter sizes and flops in your final version.

---

### Official Review · Reviewer_wjLK · 2023-11-07

**Soundness:** 2 fair
**Presentation:** 2 fair
**Contribution:** 2 fair
**Rating:** 3
**Confidence:** 4

**Summary:**

This paper focuses on the integration of weight entanglement and gradient-based methods in neural architecture search (NAS). The authors propose a scheme to adapt gradient-based methods for weight-entangled spaces, enabling an in-depth comparative assessment of the performance of gradient-based NAS in weight-entangled search spaces. The findings reveal that this integration brings forth the benefits of gradient-based methods while preserving the memory efficiency of weight-entangled spaces. Additionally, the paper discusses the insights derived from the single-stage approach in designing architectures for real-world tasks.

**Strengths:**

The paper proposes a scheme to adapt gradient-based methods for weight-entangled spaces in neural architecture search (NAS). This integration of weight-entanglement and gradient-based NAS is a new approach that has not been explored before. The paper also presents a comprehensive evaluation of the properties of single and two-stage approaches, including any-time performance, memory consumption, robustness to training fraction, and the effect of fine-tuning.

**Weaknesses:**

1. The novelty is limited. Some works have also suggested that the weights of large kernel convolution operations can be shared in differentiable neural architecture search. For example, MergeNAS: Merge Operations into One for Differentiable Architecture Search (in IJCAI20)
2. The performance improvements are limited compared with the baselines.

**Questions:**

1. How to entangle non-parameter operations, such as skip or pooling, as they have no weight compared to convolution.
2. The search cost of original DrNAS is 0.4 GPU-Days in DARTS search space. Whereas, in Table 4, the search time is 29.4 GPU-Hours. Can the authors explain the reason for the different search cost?
3. It would be beneficial to provide a more detailed explanation of the rationale behind the selection of specific search types, optimizers, and supernet types for the comparative evaluation.
4. To discuss potential limitations and challenges in implementing the proposed scheme for adapting gradient-based methods for weight-entangled spaces would provide valuable insights. This could include addressing potential constraints, trade-offs, and practical considerations in real-world implementation.

---

> ### Author Response · Authors · 2023-11-19
> **Response to reviewer wjLK**
>
> We thank the reviewer for the detailed comments and suggestions. We respond to each of your questions below:
>
> > The novelty is limited…MergeNAS
>
> We thank the reviewer for pointing out the "MergeNAS" paper, which we were unaware of at the time of writing the paper. We agree that on cell-based spaces MergeNAS works similarly to TangleNAS and we will update the manuscript to cite MergeNAS. However, MergeNAS does not apply and evaluate on MobileNet or transformer spaces ie. the weight entanglement spaces. We on the contrary extend weight entanglement to linear layers, batchnorm and layernorm layers, and attention blocks and show its effectiveness. Our work mainly aims at achieving two objectives. Firstly, making single-stage NAS methods amenable to the foundation model macro spaces (language models, vision transformers, MobileNets). This, in our opinion, is crucial to make single-stage methods more practically useful. Furthermore, our plug-and-play framework and our publicly available code makes integration and evaluation of any single-stage optimizer on these spaces easy.
>
> > Performance improvements are limited compared with the baselines
>
> We think that our improvements on ImageNet 1-k on AutoFormer-T spaces and MobileNet spaces are quite significant. Notably, on the same AutoFormer-T space, TangleNAS finds an architecture with **~ 3.8%** higher accuracy. We have now also evaluated TangleNAS on AutoFomer-S space where it achieves an improvement of  **~ 0.264%** compared to AutoFormer. Further, we have now added evaluations on the GPT-2 search space trained on OpenWebText. We transfer the pre-trained architectures (handcrafted and searched) to the Shakespeare dataset and observe that the architectures we discover outperform the larger handcrafted architecture while being efficient in terms of the number of parameters and latency.
>
> |  Architecture | Search-Type | Loss   | Perplexity | Params  | Inference Time |
> |---------------|-------------|--------|------------|---------|----------------|
> | GPT-2         | Manual      | 3.0772 | 21.69      | 123.59M | 113.3s        |
> | TangleNAS     | Automated   | **2.9038** | **18.243**     | 116.51M | 102.5s          |
>
>
> > How to entangle non-parameter operations, such as skip or pooling, as they have no weight compared to convolution?
>
> We acknowledge that non-parameter operations cannot be entangled. However, we would like to point out that in this work, we focus primarily on entanglement spaces which do not have skip connections as a choice of operation. These spaces, which are designed around foundation model architectures, are in fact far more realistic and practically useful than cell-based spaces (which usually have skip operations). Our goal through weight entanglement is to save as much of GPU memory as possible when applying single-stage NAS methods.
>
> > The search cost of original DrNAS is 0.4 GPU-Days in DARTS search space. Whereas, in Table 4, the search time is 29.4 GPU-Hours. Can the authors explain the reason for the different search cost?
>
> In our configurable framework we implement DrNAS with and without weight sharing for a unified comparison of search time and performance. The reported numbers are for the DrNAS optimizer on the DARTS search space using this unified codebase. The differences in search time are likely due to the differences in the GPU hardware setup (gpu types, cpu cores) we use in comparison to the authors.

---

> ### Author Response · Authors · 2023-11-19
>
> > It would be beneficial to provide a more detailed explanation of the rationale behind the selection of specific search types, optimizers, and supernet types for the comparative evaluation.
>
> In the single-stage neural architecture search literature [1],  it is well-known that several of the single-stage optimizers such as DARTS and GDAS have several failure modes. We started our study on weight entanglement spaces by studying DrNAS, DARTS and GDAS on the two tiny search spaces which we create benchmarks for. We find in our initial analysis that DrNAS performs competitively and is robust on both of these spaces in comparison to DARTS and GDAS. Hence we restrict ourselves to DrNAS in our large scale experiments. We have also evaluated DARTS and GDAS on the AutoFormer-T space on CIFAR-100 and presented the results below.
>
> Toy conv macro (CIFAR10)
>
> |  Optimizer | Test-acc          |
> |------------|-------------------|
> | DrNAS      | **83.02**         |
> | SPOS+RS    | 81.2525           |
> | SPOS+RE    | 81.89             |
> | DARTS_v1   | 81.61             |
> | DARTS_v2   | 81.49             |
> | GDAS       |  10% (degenerate) |
>
> Toy cell entangled (Fashion MNIST)
>
> |  Optimizer  | Test-acc  |
> |------------|-----------|
> | DrNAS      |  **90.93** |
> | DARTS-V1   | 89.905    |
> | DARTS-V2   | 90.7475   |
> | GDAS       | 90.618    |
> | SPOS+RS    |  90.6875   |
> | SPOS+RE    | 90.595    |
>
>
> AutoFormer-T (CIFAR100)
>
> |  Optimizer | CIFAR100 Test-acc  |
> |------------|---------------|
> |TangleNAS-DARTS    |    82.107 ± 0.392        |
> | TangleNAS-GDAS    |     82.12 ± 0.2813        |
> | TangleNAS-DrNAS   |   **82.668 ± 0.161**        |
> | SPOS+ES   |       82.5175 ± 0.114      |
> | SPOS+RS  |    82.210 ± 0.14242          |
>
> We choose our search spaces with three primary factors in mind (1) Architecture Diversity: ViTs, LLMs, mobilenets i.e. convolutional space (2) Application Diversity: Classification and Language Modelling) (3) Usefulness: A focus on spaces constructed around foundation model architectures
>
> [1] White, C., Safari, M., Sukthanker, R., Ru, B., Elsken, T., Zela, A., Dey, D. and Hutter, F., 2023. Neural architecture search: Insights from 1000 papers. arXiv preprint arXiv:2301.08727.
>
> > To discuss potential limitations ...
>
> We thank the reviewer for this question. In our opinion the primary limitation of our work is the inability to handle multiple architecture constraints like parameter-sizes, FLOPS directly. It is indeed possible to add a constrained secondary penalty to our loss. However , this modification still doesn’t allow one to generate a full Pareto Front of objectives which is useful in many deep learning applications (eg: fairness, robustness, hardware constraints). We leave this extension for future work.  Further, like 2-stage methods (OFA, AutoFormer), our approach also requires some amount of engineering effort to shard and combine weights into a mixture.
>
> Furthermore now we also conduct an experiment to search for models with smaller parameter sizes. We add a differentiable parameter penalty to the loss function to enforce this. We then perform the evolutionary search of SPOS setting the parameter size of the TangleNAS model as the constraint. We find that the architecture SPOS discovers in this constrained version is still worse than the TangleNAS architecture discovered. This shows the ability of our search method to discover better architectures even at smaller parameter budgets in its constrained version.
> | Optimizer | CIFAR10           | Params    | CIFAR100         | Params  |
> |-----------|-------------------|-----------|------------------|---------|
> | TangleNAS | **97.254925±0.11523** | 6.647626M | **81.25467±0.27151** | 7.08394M |
> | SPOS+RE   | 97.16425±0.13604  | 6.56041M   | 80.8324±0.23757  | 6.8866M  |
>
> If the reviewer has any follow-up questions we would be happy to address them.

---

> ### Author Response · Authors · 2023-11-21
> **Followup on pending questions**
>
> > The search cost of the original DrNAS is 0.4 GPU-Days…
>
> We would like to thank the reviewer for raising the concern about the total time taken by our method. In response, we investigated the matter further, re-ran our experiments, and  uncovered two important findings. Firstly, we misreported the total search time on the DARTS search space for DrNAS. Secondly, we had bottlenecks in our code which made our implementations slower than its original counterpart (such as the suboptimal number of workers in the data loader, and unnecessary iterations over all the modules of the model in every step of the optimization loop). Having fixed these, we are happy to report that our DrNAS with weight entanglement is now faster than the DrNAS baseline published by the authors. For a fair comparison against the authors’ baseline, we ran our code and theirs on the same hardware (RTX-2080 with 12 CPU cores), Python environment, and using the same experimental setup. Additionally, we ran experiments with and without the bottlenecks in our code fixed. The results are summarized in the table below, where WS indicates weight-sharing and WE indicates weight-entanglement.
>
> | Optimizer                   	| Time Taken (GPU days) |
> |---------------------------------|------------------|
> | DrNAS - WS (with bottleneck)	| 0.45        	|
> | DrNAS - WS (without bottleneck) | 0.38         	|
> | DrNAS - WE (with bottleneck)	| 0.42         	|
> | DrNAS - WE (without bottleneck) | 0.31         	|
> | Original DrNAS codebase             	| 0.36         	|
>
>
> Please note that DrNAS - WS (with bottleneck) and DrNAS - WE (with bottleneck) are what should have been reported in the paper. We thank the reviewer for giving us the opportunity to identify and fix this error.
>
>
>
> > Evaluation on other optimizers..
>
> We have now evaluated DARTS and GDAS on the AutoFormer-T space for CIFAR10 too. Similar to the results on CIFAR100, here too TangleNAS-DrNAS performs best out of the three choices of optimizers. This motivates our choice of gradient-based NAS optimizer in our experiments.
>
> AutoFormer-T (CIFAR10)
>
> |  Optimizer | CIFAR10 Test-acc  |
> |------------|---------------|
> |TangleNAS-DARTS    |    97.672 ± 0.04    |
> | TangleNAS-GDAS    |    97.45 ± 0.096       |
> | TangleNAS-DrNAS   |   **97.872 ± 0.054**        |
> | SPOS+ES   |       97.6425 ± 0.023      |
> | SPOS+RS  |    97.767 ± 0.024         |

---

### Meta-Review · Area_Chair_fcag · 2023-12-08

**Metareview:**

This study proposes a generalized scheme to adapt gradient-based neural architecture search methods for weight-entangled spaces. Such an integration is able to bring both the search efficiency of gradient-based methods and also the memory efficiency of weight-entangled spaces. The paper is well written and has a clear motivation. The problem to solve is an important challenge in the NAS community. Multiple benchmarks are considered for evaluation. However, multiple reviewers are concerned about the lack of novelty of the proposed method, and the insufficient comparison with related works. The AC agree that the paper is not ready for publication. More comprehensive comparison required by the reviewers should be conducted.

**Justification For Why Not Higher Score:**

Lack of novelty and insufficient experimental comparison.

**Justification For Why Not Lower Score:**

N/A

---

### Decision · Program_Chairs · 2024-01-16

Reject